# Oropharyngeal Carriage of *hpl*-Containing *Haemophilus haemolyticus* Predicts Lower Prevalence and Density of NTHi Colonisation in Healthy Adults

**DOI:** 10.3390/pathogens10050577

**Published:** 2021-05-10

**Authors:** Brianna Atto, Dale Kunde, David A. Gell, Stephen Tristram

**Affiliations:** 1School of Health Sciences, University of Tasmania, Newnham Drive, Launceston, TAS 7248, Australia; dale.kunde@utas.edu.au; 2School of Medicine, University of Tasmania, 17 Liverpool Street, Hobart, TAS 7000, Australia; david.gell@utas.edu.au

**Keywords:** nontypeable *Haemophilus influenzae*, *Haemophilus haemolyticus*, haem-binding protein, haemophilin, respiratory probiotic

## Abstract

Nontypeable *Haemophilus influenzae* (NTHi) is a major respiratory pathogen that initiates infection by colonising the upper airways. Strategies that interfere with this interaction may therefore have a clinically significant impact on the ability of NTHi to cause disease. We have previously shown that strains of the commensal bacterium *Haemophilus haemolyticus* (Hh) that produce a novel haem-binding protein, haemophilin, can prevent NTHi growth and interactions with host cells *in vitro*. We hypothesized that natural pharyngeal carriage of Hh strains with the *hpl* open reading frame (Hh-*hpl*^+^) would be associated with a lower prevalence and/or density of NTHi colonisation in healthy individuals. Oropharyngeal swabs were collected from 257 healthy adults in Australia between 2018 and 2019. Real-time PCR was used to quantitatively compare the oropharyngeal carriage load of NTHi and Hh populations with the Hh-*hpl*^+^ or Hh-*hpl*^−^ genotype. The likelihood of acquiring/maintaining NTHi colonisation status over a two- to six-month period was assessed in individuals that carried either Hh-*hpl*^−^ (*n* = 25) or Hh-*hpl*^+^ (*n* = 25). Compared to carriage of Hh-*hpl*^−^ strains, adult (18–65 years) and elderly (>65 years) participants that were colonised with Hh-*hpl*^+^ were 2.43 or 2.67 times less likely to carry NTHi in their oropharynx, respectively. Colonisation with high densities of Hh-*hpl*^+^ correlated with a low NTHi carriage load and a 2.63 times lower likelihood of acquiring/maintaining NTHi colonisation status between visits. Together with supporting *in vitro* studies, these results encourage further investigation into the potential use of Hh-*hpl*^+^ as a respiratory probiotic candidate for the prevention of NTHi infection.

## 1. Introduction

Nontypeable *Haemophilus influenzae* (NTHi) is a major bacterial cause of opportunistic infections in the respiratory tract, most notably otitis media in infants and young children, community-acquired pneumonia in the elderly, and acute exacerbations in individuals with chronic obstructive pulmonary disease (COPD) [1,2]. Collectively, these infections and subsequent long-term health complications, such as hearing loss or decline in lung function, impart a significant global disease burden [3,4]. Asymptomatic pharyngeal colonisation occurs in 20–30% of healthy adults and 20–80% of children under the age of 5 and is characterised by the simultaneous carriage of multiple strains and a rapid turnover rate of constituent genotypes [1,2,5,6,7,8]. Week to week turnover rates of NTHi genotypes as high as 62% have been reported from healthy children attending day care [9]. In the majority of cases, NTHi strains are typically replaced within three months of acquisition, but the persistent colonisation of a single strain up to six or seven months has been reported [6,9]. Although the mechanisms that lead from colonisation to infection are poorly understood, the frequency of pharyngeal colonisation, especially with immunologically new NTHi strains, has been directly linked to an increased risk of developing acute otitis media [10,11] and acute exacerbations of chronic obstructive pulmonary disease [8,12]. Higher bacterial loads in the nasopharynx have also been correlated with an increased risk of developing acute otitis media [11,13,14] and a clinically significant increase in respiratory symptoms in COPD, even in the absence of a clinical exacerbation [15]. Thus, eradicating or reducing the pharyngeal load of NTHi may have a clinically significant impact on the ability of NTHi to cause infection.

The need for alternative preventative therapies for NTHi infections has become apparent following ineffective vaccination attempts and a rapidly evolving antibiotic resistance profile that has resulted in treatment failure with first- and second-line antibiotic regimens [4,16,17,18,19,20]. One avenue gaining attention is the use of upper respiratory tract commensals, which may have utility in altering the pharyngeal habitability for pathogens such as NTHi [21]. We have previously shown that some strains of the closely related respiratory tract commensal *Haemophilus haemolyticus* (Hh) produce a haem-binding protein, haemophilin (Hpl), that has been shown to inhibit NTHi growth *in vitro* [22,23] and interactions with host cells in model respiratory cell lines [24]. NTHi-inhibitory activity in these models was speculated to be mediated by Hpl haem binding that limited NTHi access to the essential nutrient haem [22], which is required for growth, survival, and pathogenicity [25].

We hypothesised that natural pharyngeal carriage of Hh strains with the *hpl* open reading frame (Hh-*hpl*^+^) would be associated with a lower prevalence and/or density of NTHi colonisation in healthy individuals. To explore this hypothesis, real-time PCR was used to quantitatively compare the carriage load of NTHi and Hh populations with the Hh-*hpl*^+^ or Hh-*hpl^−^* genotype from the oropharynx of 257 healthy adults in Australia collected between 2018 and 2019. Carriage of Hh-*hpl*^+^ was associated with a significantly lower likelihood of concurrent NTHi carriage, long-term maintenance, or acquisition of NTHi colonisation status. Additionally, NTHi density was negatively correlated with Hh-*hpl*^+^ carriage density. This work supports further investigation into the potential use of Hh-*hpl*^+^ as a respiratory probiotic candidate for the prevention of NTHi colonisation and disease.

## 2. Results and Discussion

### 2.1. Carriage Profile of NTHi and Hh Varied between Participant Age Groups

NTHi was detected by real-time PCR of the *siaT* gene target in 29% of adult (18–65 years) participants compared to 53% of elderly (>65 years) participants (Appendix A). Conversely, the prevalence of Hh carriage was higher in adults (77%) compared to the elderly (52%) participants. Among Hh carriers, the frequency of detecting the Hh-*hpl*^+^ genotype was slightly higher in the adults (55%) than in the elderly participants (39%). Together, the lower propensity for Hh and Hh-*hpl*^+^ carriage in elderly participants may describe an environment that favours NTHi carriage.

NTHi carriage rates observed in this study are largely consistent with previous reports of nasopharyngeal carriage rates of 23–31% in healthy adults in the UK and Indigenous communities in Australia [5,26,27,28,29]. However, carriage rates as low as 3–15% have also been reported in Kenya and Nepal [30,31,32], highlighting geographical differences in NTHi carriage. Despite their predisposition to NTHi-associated infections [33], information surrounding carriage rates in elderly demographics is limited to studies in Germany (2012–2013) and Brazil (2017), reporting rates of 1.9–2.5% [1,34]. It is unclear whether the substantially higher NTHi carriage rates detected in elderly participants is representative of the local population or of sampling bias. Although the *SiaT* PCR target may also be detected in capsular *H. influenzae* [35], carriage of these strains is uncommon and collectively accounts for around 1.0% of *H. influenzae* isolates [36,37]. Thus, the false detection rate of NTHi strains is likely to be extremely low and unlikely to affect statistical findings presented in this study. This does not compromise the clinical utility of Hh-*hpl*^+^, as Hpl also exhibits inhibitory activity against capsular strains *in vitro* [22]. This is the first study to assess Hh carriage prevalence among adults at a community level.

In addition to geographical and age-related variances, NTHi carriage rates vary considerably between studies, largely owing to the different culture- and molecular-based methods employed and the difficulty of distinguishing NTHi from Hh [38,39]. Although the nasopharynx is the preferred collection site for *H. influenzae* isolation in culture-based carriage studies [38], several studies have reported similar or improved detection of NTHi and Hh from oropharyngeal (OP) collections using qPCR-based methods [27,40] that can reliably differentiate the two species, such as the *hypD* and *siaT* targets employed in this study [35]. Therefore, collection site is unlikely to impact carriage rates determined in this study.

### 2.2. Carriage of hpl-Positive Hh Is Correlated with Reduced Prevalence and Density of NTHi Cocolonisation

Elderly participants were 2.43 times (95% CI, 1.95–2.61; *p* < 0.0001) and adult participants were 2.67 times (95% CI, 2.63–2.70; *p* = 0.0036) less likely to carry NTHi if Hh-*hpl*^+^ strains were detected, compared to the carriage of Hh-*hpl^−^* strains. NTHi carriage prevalence was highest in adult (62%) and elderly (91%) participants that concurrently carried Hh strains that did not possess the *hpl* ORF (Hh-*hpl*^−^) in their oropharynx. Among participants carrying Hh-*hpl*^+^ strains, NTHi carriage rates were 25% (adults) and 14% (elderly) or 13% (adults) and 0% (elderly) in participants where Hh-*hpl*^+^ represented the predominant Hh genotype (Figure 1A).

Comparison of Hh densities determined by qPCR of the *hpl* and *hypD* gene targets suggested concurrent carriage of multiple Hh genotypes, where not all possessed the *hpl* ORF. Simultaneous colonisation with multiple Hh and NTHi genotypes has previously been described in a longitudinal study of healthy adults [6]. Hh-*hpl*^+^ represented the predominating genotype in 49% (39/79) and 53% (8/15) of cases when the *hpl* gene was detected in adult and elderly participants, respectively (Appendix A). The proportionate density of Hh-*hpl*^+^ (as a function of total Hh carriage) was negatively correlated with NTHi density among adult (*r_s_* = −0.16; 95% CI, −0.314–−0.006; *p* = 0.0366) and, to a larger degree, elderly (*r_s_* = −0.53; 95% CI, −0.7106–0.2851; *p* < 0.0001) participants that carried either species. In the adult age group, the average proportion of NTHi density decreased by 20% among individuals who concurrently carried Hh-*hpl*^+^ as the non-predominant Hh genotype or by 47% if Hh-*hpl*^+^ represented the predominant Hh genotype. This trend was more pronounced in the elderly age group where the average proportion of NTHi density decreased by 83% among individuals who concurrently carried Hh-*hpl*^+^ or by 88% if Hh-*hpl*^+^ represented the predominant Hh genotype (Figure 1B). Together these data suggest that carriage of Hh-*hpl*^+^, but not Hh strains lacking the *hpl* ORF, lowers the incidence and density of concurrent NTHi carriage, particularly if Hh-*hpl*^+^ represents the predominant Hh genotype. 

### 2.3. Carriage of hpl-Positive Hh Prevents Persistent Colonisation or Acquisition of NTHi Carriage Status

To investigate the risk of acquiring/maintaining NTHi colonisation status, follow-up OP swabs were collected from individuals that carried *hpl*^+^ (*n* = 25) or *hpl^−^* (*n* = 25) strains of Hh. At visit 2, Hh-*hpl*^+^ colonisation status was maintained in 19/25 of individuals, with an additional 11 participants gaining colonisation status, resulting in a total Hh-*hpl*^+^ carriage rate of 60% (30/50) at visit 2. Maintenance or acquisition of NTHi carriage status at visit 2 was associated with the carriage of Hh-*hpl*^−^ strains at visit 1 and at visit 2 in 75% (12/16) of cases (Table 1). The remaining four NTHi carriers were co-colonised with Hh-*hpl*^+^ strains; however, in all cases, Hh-*hpl*^+^ was not the predominant Hh genotype. In contrast, of participants who were not colonised with NTHi (either by loss of NTHi or who were never colonised), 88% (30/36) were carrying Hh-*hpl*^+^ (Table 1). The likelihood of being colonised with NTHi at visit 2 (either by maintaining or acquiring NTHi colonisation status) was 2.63 times (95% CI, 2.56–2.70, *p* = 0.0112) lower in individuals colonised with Hh-*hpl*^+^ strains at either visit 1 or visit 2. These results suggest that Hh-*hpl*^+^ colonisation may have a protective effect against NTHi colonisation *in vivo*. However, the data can only account for total changes in carriage status between two visits and cannot assess the characteristically diverse and dynamic turnover of individual NTHi/Hh genotypes. Therefore, these findings may underestimate the protective capacity of Hh-*hpl*^+^ carriage and additional longitudinal studies with genotypic resolution are warranted. The findings presented by this study are only correlative and, as such, cannot rule out other unmeasured host-derived factors that may affect an individual’s susceptibility to NTHi colonisation, such as smoking, airway dysbiosis, and underlying chronic respiratory diseases [41,42,43,44].

### 2.4. Potential Therapeutic Utility of hpl-Positive Strains of Hh

In summary, the carriage of Hh-*hpl*^+^ was associated with a significantly lower proportionate density and prevalence of concurrent NTHi carriage and long-term maintenance or acquisition of NTHi colonisation status. These findings suggest a potential protective role of Hh-*hpl*^+^ strains against NTHi pharyngeal colonisation, particularly in an elderly population with a predisposition to NTHi colonisation in the absence of Hh-*hpl*^+^. The frequency and density of NTHi colonisation or the acquisition of immunologically new strains are factors associated with an increased risk of disease onset and severity [8,12]. Thus, a commensal bacterium that can prevent NTHi pharyngeal colonisation incidence and/or bacterial load has compounded therapeutic utility and may be favourable over immunogenic approaches that are hampered by the highly variable expression of NTHi surface proteins and immunogenicity that does not protect against reinfection [45]. However, clinical trials are required to determine if Hh-*hpl*^+^ can eradicate or protect against direct NTHi challenge *in vivo*, particularly in populations predisposed to NTHi infections.

Production of the Hpl haemophore has previously been shown to mediate the *in vitro* inhibitory capacity of Hh against NTHi growth and the adherence/invasion of model respiratory cell lines by restricting NTHi access to the essential nutrient haem [22,23,24]. Although we postulate that the same mechanism may be involved here, the study design reports on the presence of the *hpl* coding region and does not assess phenotypic production of the Hpl protein. We have previously shown that even among Hh strains containing identical *hpl* ORF sequences, some strains lack the capacity to express *hpl* and produce the Hpl protein that mediates NTHi inhibitory activity *in vitro* [23,24,44]. However, the majority (16/17) of Hh-*hpl*^+^ detectable by our PCR assay from our culture collection produce Hpl and elicit NTHi-inhibitory activity (Appendix A), and there are no incidences of Hpl production in strains that do not contain the *hpl* ORF. Further work is underway to determine the genetic determinants of *hpl* expression and Hpl production. The involvement of other host-mediated responses or interactions with other microbial communities in the oropharynx also cannot be excluded. Immune modulation, rather than physical competition, was shown to play an important role in the protective capacity of intranasal *Muribacter muris* (Hh surrogate) against NTHi colonisation and infection in a murine NTHi otitis media model [46]. Additionally, other microbial upper respiratory tract commensals capable of producing bacteriocins against common pathogens have been reported [47,48]. However, the substantial effect sizes correlating Hh-*hpl*^+^ and NTHi density despite potential confounders support a protective role of Hh-*hpl*^+^ against NTHi pharyngeal colonisation in healthy adults.

The *in vivo* correlations from the current study, together with previously published causative *in vitro* evidence, support further investigation into the potential use of *hpl*-positive strains of Hh as a respiratory probiotic candidate for the prevention of NTHi colonisation and disease.

## 3. Materials and Methods

### 3.1. Study Population

Participants (*n* = 257) were comprised of community groups and university staff and students in Tasmania, Australia. Recruitment and sample collection was conducted between June 2018 and November 2019. All participants were briefed on study details and received written information prior to giving informed consent to participate. Participants were included in the study if they fulfilled the following criteria: (i) over 18 years of age, (ii) not currently taking antibiotics, and (iii) not experiencing respiratory-related symptoms. The study was approved by the Tasmanian Health and Medical Human Research Ethics Committee (Ref No: H0016835, approved 11 December 2017) in accordance with the National Statement on the Ethical Conduct in Human Research (NHMRC 2007, updated 2014).

### 3.2. Oropharyngeal Swab Collection

Oropharyngeal (OP) swabs were collected due to ease of collection and participant tolerance. Several studies have reported similar or improved qPCR detection of NTHi and Hh from OP swabs compared to nasopharyngeal swabs [1,27,38,40]. OP swabs were collected by two investigators by depressing the tongue and rolling the tip of a sterile cotton swab on the posterior wall of either side of the oropharynx for 2 seconds, avoiding contact with the surface of the mouth to minimise contamination with mouth flora. Follow-up swabs were collected 2–6 months following the initial visit from a randomly selected subset of this population (all ages) that carried either Hh-*hpl^−^* (*n* = 25) or Hh-*hpl*^+^ (*n* = 25) at the first visit.

Immediately following collection, swabs were stored in 1 mL of room temperature transport media containing skim milk, tryptone, glucose, and glycerin (STGG) and transported to the laboratory within 2 h. STGG has previously been described and evaluated for optimal storage [49] and PCR detection of *Haemophilus influenzae* from nasopharyngeal swabs [50]. Specimens were subject to a vigorous vortex for 1 min to disperse organisms from the swab tip, and the tip was removed from the media by pressing the swab against the wall. Media suspensions were frozen at −80 °C until analysis.

### 3.3. Real-Time PCR Quantification of NTHi, Hh, and Hh-hpl^+^

Template gDNA was prepared from thawed 500 µL OP suspension aliquots using the DNeasy Blood & Tissue kit (Qiagen) following the standard proteinase K extraction protocol. NTHi and Hh strains containing the *hpl* ORF were simultaneously detected and quantified from OP swab gDNA by using a previously optimised and validated triplex real-time PCR assay [23]. Briefly, this PCR assay adapted the use of previously validated genes for the discrimination of Hh (*hypD*) and *H. influenzae* (*siaT*) [35] as well as primers specific for the detection of the *hpl* ORF (GenBank MN720274) in Hh strains. This assay was validated for the detection of *hpl* sequences within 85–100% similarity to the PCR amplicon, which accounts for all known Hpl-producing (and thus NTHi-inhibitory) isolates. Hh strains with more divergent sequences have been isolated; however, none are known to produce Hpl based on bioactivity assays (data not shown). Further details of PCR validation, including the limits of detection and quantification of gene targets, are supplied in the Appendix A. PCRs were performed using the CFX96 Touch^TM^ real-time PCR system (Bio-Rad) in 96-well optical plates. The complete details of PCR thermocycling conditions, reagents, primer/probe design, and assay optimisation/validation have previously been described [23]. Each run included a negative control (*H. parainfluenzae* ATCC 7901), no template control, and 10-fold dilutions of a standard containing 2 × 10^−8^ ng NTHi ATCC 49247 and Hh-BW1 (known Hh-*hpl*^+^ strain) gDNA. The quantification of NTHi and Hh was expressed as genome equivalents calculated from the appropriate standard, as previously described [35]. Bacterial carriage status was considered negative for Ct values above 35 or if genome equivalents fell below the limit of quantification for the corresponding gene target. The Hh-*hpl*^+^ genotype was considered predominant if genome equivalents calculated from the *hpl* target accounted for more than 50% of the total calculated Hh (from the *hypD* target).

### 3.4. Statistical Analysis

Based on reported age-associated variation in the pharyngeal carriage rate and density of NTHi, analyses were stratified by age (18–65 years and >65 years). Logistic regression models were used to assess whether Hh carriage status (either absolute presence of Hh-*hpl*^+^ or Hh-*hpl^−^*) or age predicted NTHi colonisation by measuring odds ratios (ORs) for each bacterium associated with the density of the other species. A simple logistic regression was also used to measure the OR between Hh-*hpl*^+^ carriage and NTHi colonization at follow up. A nonparametric Spearman correlation was conducted to determine the correlation between the proportion Hh-*hpl*^+^ carriage density (as a function of total Hh) and the proportion of NTHi carriage in the swabs.

## Figures and Tables

**Figure 1 pathogens-10-00577-f001:**
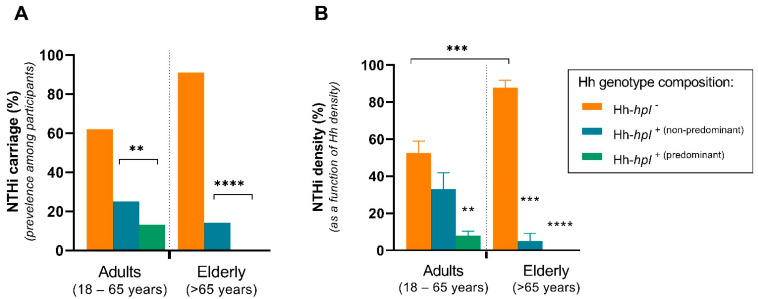
NTHi dominance in oropharyngeal swabs of healthy adult (18–65 years) or elderly (>65 years) participants co-colonised with Hh. NTHi oropharyngeal carriage prevalence (**A**) or proportion of NTHi (as a function of total Hh) (**B**) among participants concurrently carrying Hh strains that possess the *hpl* ORF (Hh-*hpl*^+^) or do not possess the *hpl* ORF (Hh-*hpl^−^*). Hh-*hpl*^+^ (predominant) denotes instances where *hpl*^+^ is the predominant Hh genotype (>0.5 of total Hh). Error bars represent ±SEM; statistical significance was determined by simple logistic regression **(A)** or nonparametric Spearman correlation (**B**); ** *p* < 0.005, *** *p* < 0.001, **** *p* < 0.0001.

**Table 1 pathogens-10-00577-t001:** NTHi colonisation status in participants between visit 1 and visit 2 (*n* = 50).

	NTHi Colonisation at Visit 1 -> Visit 2 Frequency (%)			
Hh Genotype	NTHi+ | NTHi− (Colonisation Loss)	NTHi− | NTHi− (Never Colonised)	NTHi+ | NTHi+ (Consistently Colonised)	NTHi− | NTHi+ (Colonisation Gain)
**Total**	8/50 (16)	26/50 (52)	4/50 (8)	12/50 (24)
***hpl*^+^**	7/8 (88)	23/26 (88)	2/4 (50)	2/12 (17)
***hpl^−^***	1/8 (12)	3/26 (12)	2/4 (50)	10/12 (83)

^+/−^ Detection by PCR for corresponding gene targets *siaT* (NTHi), *hypD* (Hh) and the *hpl* ORF.

## Data Availability

The data presented in this study are contained within the article or Appendix A.

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
