# Peer review of "Oropharyngeal Carriage of hpl-Containing Haemophilus haemolyticus Predicts Lower Prevalence and Density of NTHi Colonisation in Healthy Adults"

_pathogens, 2021, doi:10.3390/pathogens10050577_

Round 1
Reviewer 1 Report
Oropharyngeal carriage of hpl-containing Haemophilus haemolyticus predicts lower prevalence and density of NTHi colonisation in healthy adults
Pathogens 1199647
Summary: This study nicely demonstrates the correlation of Haemophilus haemolyticus oropharyngeal colonization with reduced frequency and density of NTHi colonization. The techniques are clever and solid. The beauty of this study is its simplicity and potential importance in further understanding the complex relationships of Haemophili
that colonize the human oropharynx.
Major concerns: None
Minor concerns:
The first sentence in the introduction is incorrect. Even prior to introduction of the conjugate type b vaccine, colonization with type b (or any of the cap
- types) was fairly uncommon, except in contacts of children with type b infection. One of the best references for this is: Non-typeable Haemophilus influenzae, an under-recognised pathogen. Hyg., Camb. (1963), 61, 247 D.C. Turk
- This reviewer had trouble making the numbers of table S1 work on initial reading. To make the table more reader friendly, consider indicating that the Hh-hpl- and Hh-hpl+ strains are subsets of the Hh+ isolates (maybe placing the NTHi+ strains at the top of the table, followed by the Hh- strains, and then the Hh+ strains, with the Hh-Hpl strains indented under the Hh+ strains).
- It’s not clear what the 94% in lines 192-4 refers to, except maybe 23/24 strains tested (the 24th strain that presumably didn’t produce Hpl isn’t listed on table S3) and the sentence is misleading, because 8/24 test strains didn’t produce Hpl (interestingly, those with the lowest % identity of their hpi) ORFs).
- Another possible intervention against NTHi colonization (and one more IRB-friendly) would be topical application of an iron or heme chelater.
Author Response
The authors would like to thank the reviewers for their time and constructive feedback. Our responses to comments/suggestions have been noted below in red, and any changes to the manuscript have been highlighted.
Major concerns: None
Minor concerns:
- The first sentence in the introduction is incorrect. Even prior to introduction of the conjugate type b vaccine, colonization with type b (or any of the cap types) was fairly uncommon, except in contacts of children with type b infection. One of the best references for this is: Non-typeable Haemophilus influenzae, an under-recognised pathogen. , Camb. (1963), 61, 247 D.C. Turk
We have removed this sentence.
- This reviewer had trouble making the numbers of table S1 work on initial reading. To make the table more reader friendly, consider indicating that the Hh-hpl- and Hh-hpl+ strains are subsets of the Hh+ isolates (maybe placing the NTHi+ strains at the top of the table, followed by the Hh- strains, and then the Hh+ strains, with the Hh-Hpl strains indented under the Hh+ strains).
The table has been adjusted to the reviewer’s suggestion.
- It’s not clear what the 94% in lines 192-4 refers to, except maybe 23/24 strains tested (the 24th strain that presumably didn’t produce Hpl isn’t listed on table S3) and the sentence is misleading, because 8/24 test strains didn’t produce Hpl (interestingly, those with the lowest % identity of their hpi) ORFs).
Our PCR assay is validated to detect hpl sequences with 85-100% similarity to the BW1 strain PCR amplicon (see line 244-245). Of the total 24 strains in Table S4, 17 are detected by the triplex PCR assay but only 16 of these are capable of producing Hpl. This means that there is a high likelihood of the strains we detect from throat swabs being phenotypic producers of Hpl. We have changed the 94% to “16/17”, and indicated these strains in Table S4, to make this clearer.
- Another possible intervention against NTHi colonization (and one more IRB-friendly) would be topical application of an iron or heme chelater.
We thank the reviewer for this suggestion about a potential clinical utility of haem/iron chelation against NTHi and will consider this in future discussions. No changes have been introduced to the current manuscript.
Reviewer 2 Report
Atto et al evaluated their hypothesis that the commensal Hh hpl+ prevented HTHi colonization in vivo as well as in vitro. The results were very interesting.
The manuscript was overall well-written.
I have a few minor suggestions.
- Please provide the detail of participants of the follow-up study. How did the authors select the participants? age population, area…. and so on. Adding a figure to explain the relationship would be helpful.
- Please provide the detailed criteria (such as cutoff value) of real-time PCR to judge the carriage or not.
Author Response
The authors would like to thank the reviewers for their time and constructive feedback. Our responses to comments/suggestions have been noted below in red, and any changes to the manuscript have been highlighted.
- Please provide the detail of participants of the follow-up study. How did the authors select the participants? age population, area…. and so on. Adding a figure to explain the relationship would be helpful.
Participants were randomly selected from the same population group for follow-up swabs. The only criteria for their selection was that they carried Hh (either hpl- or hpl+) at visit 1. Line 226-7has been edited to emphasize the origin of this sample group to read “from a randomly selected subset of the population (all ages) that carried either Hh-hpl- (n=25) or Hh-hpl+ (n=25) on the first visit”
- Please provide the detailed criteria (such as cutoff value) of real-time PCR to judge the carriage or not.
Criteria for a positive/negative interpretation has been added to the methods (see line 256-258).
Round 2
Reviewer 1 Report
No concerns about this nice revision.
Reviewer 2 Report
The authors have addressed all my concerns. This paper is now ready to read for the scientific community.